# Investigating Scottish Long COVID community rehabilitation service models from the perspectives of people living with Long COVID and healthcare professionals: a qualitative descriptive study

Edward Duncan ,[1] Lyndsay Alexander ,[2,3] Julie Cowie ,[4] Alison Love,[5] Jacqui H Morris ,[6] Rachel Moss,[7] Jane Ormerod,[5] Jenny Preston,[8] Joanna Shim ,[3] Emma Stage ,[3] Tricia Tooman,[1] Kay Cooper [3]

For numbered affiliations see end of article.

**Correspondence to**
Professor Edward Duncan;
edward.duncan@stir.ac.uk

## ABSTRACT

**Objectives** This study aimed to explore the perceptions and experiences of barriers and facilitators to accessing Long COVID community rehabilitation.

**Design** We used a qualitative descriptive design over two rounds of data collection with three participant groups: (1) people with experience of rehabilitation for Long COVID (PwLC); (2) National Health Service (NHS) staff delivering and/or managing community rehabilitation services (allied health professionals (AHPs)) and (3) NHS staff involved in strategic planning around Long COVID in their health board (Long COVID leads).

**Setting** Four NHS Scotland territorial health boards.

**Participants** 51 interviews: eight Long COVID leads (11 interviews); 15 AHPs (25 interviews) and 15 PwLC (15 interviews).

**Results** Three key themes were identified: (1) accessing care for PwLC, (2) understanding Long COVID and its management and (3) strengths and limitations of existing Long COVID rehabilitation services.

**Conclusions** Organisational delivery of Long COVID community rehabilitation is complex and presents multiple challenges. In addition, access to Long COVID community rehabilitation can be challenging. When accessed, these services are valued by PwLC but require adequate planning, publicity and resource. The findings presented here can be used by those developing and delivering services for people with Long COVID.

## INTRODUCTION
### Background

Long COVID, a condition defined as ongoing COVID-19 symptoms that continue beyond twelve weeks following an initial acute COVID-19 infection, was first described in Spring 2020.[1] It is estimated that at least 65 million people globally have Long Covid.[2] By January 2023, over 2 million people were

## STRENGTHS AND LIMITATIONS OF THIS STUDY

⇒ Studying four health boards enabled an analysis of development of differing service models of delivery over the same time period.
⇒ The rapidly evolving nature of Long COVID and its management resulted in fewer distinct differences between the four health boards than originally anticipated.
⇒ Study findings are limited to Scottish Long Covid rehabilitation context.
⇒ Other relevant perspectives, such as general practice/family medicine are not included in this study but will be presented in subsequent publications.

estimated to have Long COVID in the UK, including over 175 000 living in Scotland.[3] Long COVID has broad multiple system presentation and can have profound physical, emotional, social and financial consequences. Moreover, debilitating symptoms of Long COVID often persist for a year or more.[2]

National Health Service (NHS) funding to support the diagnosis, treatment and rehabilitation for Long COVID varies across the UK. Since the autumn of 2020, NHS England has allocated £94 million (£1.66 per capita, based on the 2021 UK Office of National Statistics Census data England's population was 56 489 800) to be invested in specialist Long COVID clinics in England to complement existing primary, community and rehabilitation care services.[4] The Welsh Government has allocated £18.3 million since 2020 (£5.88 per capita based on a population of 3 107 500 in 2021) to support the delivery of Long COVID diagnosis, treatment and rehabilitation

BMJ

within existing services.[5] In September 2021, the Scottish Government announced £10 million (£1.82 per capita based on a population of 5 479 900 in 2021) over 3 years to support Scottish NHS health boards' response to Long COVID.[6] At the end of May 2022, plans for the first £3 million of that funding were announced.

The National Institute for Health and Care Excellence (NICE)[1] and the WHO[7] recommend people with Long COVID, following an initial medical assessment to exclude underlying conditions, have access to rehabilitation to aid their recovery. As with other long-term conditions, community rehabilitation for people with Long COVID should be personalised, multidisciplinary and comprehensive in order to maximise function, quality of life and participation in society.[8 9]

However, the optimal approach to deliver Long COVID community rehabilitation is currently unknown. NICE clinical guidance suggests broad rehabilitation approaches, which can be delivered in isolation or in combination, such as self-management and multidisciplinary rehabilitation.[1] The effectiveness of these approaches is not yet clear and there are further uncertainties about the most appropriate means of organising service delivery, particularly that of multidisciplinary rehabilitation. There are well-documented barriers to accessing healthcare services for people with experience of rehabilitation for Long COVID (PwLC)[10–12]; however, previous research has not specifically considered rehabilitation services. Our 2019 national survey[13] of Scotland's 14 territorial health boards found that Long COVID community rehabilitation was universally available across Scotland. At the time of the survey, one health board was delivering Long COVID community rehabilitation as a dedicated service, with the remainder integrating rehabilitation within existing services. And yet, there was substantial variation across health boards in the mode of service delivery, given that the means of optimally delivering community rehabilitation to this population was unknown.

To date, rehabilitation services for Long COVID have varied internationally with respect to delivery and the healthcare professionals involved,[14] including virtual rehabilitation focusing on self-management and education,[15] group-based pulmonary rehabilitation,[16] tiered multidisciplinary pathways[17] and phased approaches to managing symptoms.[18] Our research group are investigating models of Long COVID rehabilitation service delivery in an overarching study called LOCO-RISE. LOCO-RISE employed a realist evaluation design to investigate different service delivery models for Long COVID community rehabilitation in Scotland between November 2021 and July 2023. LOCO-RISE aims to provide responsive evidence-based recommendations to the NHS about how to most effectively deliver community rehabilitation services for people with Long COVID. The aim of the phase of LOCO-RISE reported in this paper was to explore key stakeholders perceptions and experiences of the barriers and facilitators to implementing Long COVID community rehabilitation.

## METHODS
### Study design
This paper reports on the first two rounds of data collection, which took place over a 12-month period. We employed a longitudinal qualitative descriptive[19] research design based on 51 interviews with three key participant groups, each of which provided distinct perspectives: (1) PwLC, who provided lived experiences of receiving Long COVID rehabilitation; (2) NHS staff who provided perspectives of delivering and/or managing community rehabilitation services for PwLC (allied health professionals (AHPs)); and (3) NHS staff involved in strategic planning around Long COVID in their health board (Long COVID leads) who provided broader organisational perspectives. The study is reported in accordance with the Consolidated criteria for Reporting Qualitative research (online supplemental file 1).[20] Qualitative description was selected over other qualitative designs as it enabled the exploration of a range of shared experiences to generate a rich description[19] of barriers and facilitators to the implementation of Long COVID rehabilitation.

### Setting
The study was conducted in four Scottish health boards (HB1–HB4). The boards were selected for their geographical and demographic spread as well as variation in Long COVID rehabilitation service models, summarised in table 1. As shown in table 1, all boards adopted some form of multidisciplinary team (MDT) approach using a blended mode of delivery (face to face, telephone, near me). Staffing varied; occupational therapy and physiotherapy were most consistent across services, with some including additional professions such as dietetics, speech and language therapy and psychology. Some staff were co-located, while others worked as part of a team, but geographically remote. Notably, none of the services included medical staff.

### Recruitment and data collection
Two rounds of data collection took place: round 1 (November 2021–April 2022) and round 2 (May–October 2022). In each round, three participant groups took part: PwLC, AHPs and Long COVID leads. Long COVID leads were purposively sampled by email invitation or word of mouth, for their role in leading on Long COVID within their health board, and included AHP directors, medical directors and clinical service leads. Long COVID leads who only managed in-patient services were ineligible, since this study was exploring community rehabilitation. AHPs, most often occupational therapists and physiotherapists, were involved in providing Long COVID rehabilitation. Most AHPs acted in a team member role, though a small number were also clinical service leads. AHPs involved in the treatment of PwLC in the study health board areas were purposively recruited by email invitation. The invitation was sent by senior managers on behalf of the study team and explained the reasons for the research. Student AHPs were ineligible, as were AHPs who only provided

**Table 1** Description of participating health boards

| Health board | Health board 1 | Health board 2 | Health board 3 | Health board 4 |
|---|---|---|---|---|
| Setting | Largely rural | Large urban/rural area | Largely urban | Large urban/rural area |
| Round 1 Long COVID service delivery model | Integrated service: community rapid response rehabilitation | Integrated service: community rehabilitation | Integrated service: community rehabilitation | Dedicated Long COVID service |
| | MDT: Physiotherapy, occupational therapy, nursing, community psychiatric nurse, pharmacy, advanced nurse practitioner and speech and language therapy, with links to GP and dietetics | MDT: Physiotherapy, occupational therapy, dietetics and speech and language therapy. | MDT: Physiotherapy, occupational therapy, psychology, speech and language therapy, and rehabilitation assistant practitioner, physiotherapy, occupational therapy, psychology, dietetics, and speech and language therapy | MDT: Physiotherapy, occupational therapy, psychology, dietetics, and speech and language therapy |
| | Blended delivery | Blended delivery with centralised point of referral | Blended delivery | Blended delivery |
| Round 2 Long COVID service delivery model | As round 1 | As round 1 | Integrated service moving towards a soft launch of dedicated Long COVID Service. Blended delivery with single access point | Dedicated Long COVID Service via a three-tiered triage system. Service temporarily halted due to large volume of referrals and subsequent decline in staffing and funding |

GP, general practitioner; MDT, multidisciplinary team.

in-patient Long COVID rehabilitation. Finally, a convenience sample of PwLC was recruited by their AHP. People with Long COVID were provided with a study pack in printed or electronic format that explained the purpose of the study and volunteers opted into the study by contacting the research team by email, telephone or leaving their contact details after completing a survey of health-related outcomes as part of the larger study. People with Long COVID were ineligible if they were aged under 18, required specialist (in-patient) management of Long COVID complications, had another life-limiting illness with reduced life-expectancy (eg, disseminated malignancy) or were unable or unwilling to provide informed consent. Purposive sampling criteria were informed by prior knowledge of the emerging Long COVID services in each health board. We aimed to recruit 12 AHPs, 4–8 Long COVID leads and up to 32 PwLC for each round of data collection. These sample sizes were based on pragmatic considerations including the researchers' knowledge of each included health board's community rehabilitation services staffing profiles; time and financial constraints; and the challenges of recruiting PwLC whose health limiting condition was hypothesised to make participation in the research challenging. We did not seek to reach data saturation, but instead considered the 'information power' of the sample.[21] Malterud et al's

construct of 'information power'[21] argues that it is the amount of information the sample holds, rather than the size of the sample itself, that is, the important factor when considering sample sufficiency. The 'information power' of the sample in this study was strengthened by the match of the sample to the study's specific aim, the specificity of the sample in relation to their experiences of Long COVID rehabilitation, the depth of the interviews being conducted and the applied nature of the investigation.

Interview topic guides (online supplemental file 2) were developed by the research team that included people with lived experience of Long COVID. Topic guides were further refined for round 2 to take account of the temporal nature of the questions being asked. The topic guides combined a theory-based approach using the Template for Intervention Description and Replication (TIDieR) intervention description categories[22] to enable a comprehensive description of each rehabilitation service model, which can be conceived of as an organisational level intervention. Questions to Long COVID leads were based on constructs from the Consolidated Framework for Implementation Research[23] to ascertain the factors that were associated with the intervention's implementation and a series of questions exploring facilitators and barriers to delivering and receiving community Long COVID rehabilitation. The interview topic guides were pilot tested

with people who had lived experienced of Long Covid and with two AHPs who were not involved in the study prior to implementation. Web-based (Microsoft Teams) or telephone interviews were conducted by five experienced healthcare researchers with master's or doctoral level qualifications and experience of qualitative research (VB, JC, RM, JS and TT). All interviewers were female and had experience of qualitative NHS data collection and analysis. Two of the researchers were AHPs and had prior awareness of the challenges of delivering rehabilitation. All interviewers were provided with training and supervision from KC and ED and had no prior relationship with the services or participants. Informed consent was obtained from all participants and recorded verbally prior to each interview. Interviews lasted between 22 and 86 min (average 52 min). Only the participant and interviewer were present during each interview. Interviews were audiorecorded and transcribed intelligently by an external transcription service, where redundant words and/or sounds were removed. Field notes were taken during some interviews. These notes added to team discussions to support understanding of the data as it was collected but did not form part of the official data analysis. Transcripts were not returned to participants for comment or correction.

### Data analysis
Data were analysed using the framework method, which is commonly used in applied health research and recognised as appropriate for multidisciplinary research teams.[24] As a form of thematic analysis, the framework approach was congruent with the qualitative descriptive design being employed.[19]

For each participant group, at least two researchers from the team familiarised themselves with at least 20% of transcripts, making analytical notes directly on the transcripts. The researcher pairs were from different disciplinary backgrounds (occupational therapy, physiotherapy, psychology, applied health research), thereby ensuring interpretation from a range of perspectives. Following discussion between all team members involved in familiarisation, a framework for analysis and interpretation was constructed.[24] The use of multiple researchers at this stage aimed to mitigate for bias in interpretation that may have occurred due to the disciplinary backgrounds of some team members. Although line-by-line coding is common, it is also possible to develop a framework without engaging in explicit coding[25]; we adopted the latter approach. The 'analytical framework'[23] was applied across all interview transcripts using highlighting and the comments function in Microsoft Word, with regular review and discussion within the team. A charting matrix was created in Microsoft Excel that involved summarising broad categories of data from the transcripts, then interpreting the charted data by exploring within and between-cases to derive concepts and themes. These were then grouped into overarching themes. All researchers

| Table 2 | Participant demographics | | | | |
|---|---|---|---|---|---|
| | **Long COVID leads (L)** | | **Staff delivering service/AHPs (S)** | | **Patients (P)** |
| | N=6<br>Rounds 1 and 2 N=3<br>Round 1 N=2<br>Round 2 N=1 | | N=15<br>Rounds 1 and 2 N=9<br>Round 1 N=2<br>Round 2 N=4 | | N=15 |
| Age | | | | | |
| Working age (%) | 8 (100) | | 15 (100) | | 13 (87) |
| | | | AfC Band | N= | |
| | | | 6 | 7 | |
| | | | 7 | 4 | |
| | | | 8a | 4 | |
| Sex | | | | | |
| Female (%) | 6 (75) | | 15 (100) | | 12 (80) |
| Male (%) | 2 (25) | | 0 | | 3 (20) |
| Employment status | | | | | |
| Working full-time (%) | 8 (100) | | 15 (100) | | 6 (40) |
| Working part-time (%) | 0 | | 0 | | 1 (7) |
| Phased return to work (%) | 0 | | 0 | | 2 (13) |
| Retired (%) | 0 | | 0 | | 2 (13) |
| Unable to work because of illness (%) | 0 | | 0 | | 3 (20) |
| Not provided (%) | 0 | | 0 | | 1 (7) |
| Time from first COVID diagnosis | NA | | NA | | 12–28 months |

AfC, agenda for change; AHPs, allied health professionals.

were involved in interpreting the data during regular team discussions. Participants did not directly provide feedback on the findings. However, the findings were presented in a webinar attended by Long Covid Leads, AHPs and PwLC, several of whom had participated in the study. The webinar attendees endorsed the study findings and reflected that they resonated with their personal experiences.

## Patient and public involvement
Two people with lived experience of Long COVID were core members of the research team. They co-developed study materials and interview topic guides and contributed to analysis and interpretation of study findings.

## RESULTS
### Participant characteristics
We recruited eight Long COVID leads, who took part in 11 interviews over the two rounds of data collection: 15 AHPs (25 interviews) and 15 PwLC (15 interviews). We did not meet our recruitment target for PwLC despite a high conversion rate (invitation to take part in the study resulting in consent and participation). The low recruitment rate occurred as a result of apparent barriers to PwLC being referred to the community rehabilitation services, a phenomenon that was subsequently explored in a related study (to be reported elsewhere). See table 2 for participant details. No participants who consented withdrew from the study.

### Findings
Framework analysis resulted in three key themes being identified: (1) accessing care for PwLC, (2) understanding Long COVID and its management and (3) strengths and limitations of existing Long COVID rehabilitation services. Table 3 illustrates the three themes and the categories of data that contributed to each.

Participant's data were coded and reported as follows: (1) role of participant: L=Long Covid leads; P=PwLC; S=AHPs; (2) health board (HB) number: HBs 1–4; (3) round of data collection: 1 or 2 and (4) participant number (eg, the participant identifier for the first quote: P-3-2-08 is a PwLC from HB3, data were collected in round 2 and was the eighth PwLC interviewed).

### Accessing care for people with Long COVID
Barriers to accessing Long COVID rehabilitation were identified by all participant groups. In Round 1, patients commonly attributed this to limited availability of face-to-face general practitioner (GP) appointments due to COVID-19 restrictions.

*'You don't get face-to-face with the GP. So again, when you get a GP, it's always a different GP you're talking to and you try your best, but you get locum, and they've all been fantastic. I'm not, again, I wouldn't criticise them, but it's exhausting telling the same story every single time.'* [P3208]

Long COVID leads and AHPs observed people's reluctance to seek help from services widely acknowledged as being under pressure. By round 2, they felt that the perception of GPs being closed, and services overwhelmed, needed to be questioned, and called for greater public awareness of Long COVID rehabilitation services and how to access them.

| Table 3 Key themes and categories of data | |
|---|---|
| **Key themes** | **Subthemes** |
| Accessing care for people with Long COVID | PwLC accessing general practitioner (GP) services (L; P) |
| | GP referring PwLC to community rehabilitation (L; P; S) |
| | Awareness of services by GPs and PwLC (S) |
| | Signposting (S) |
| | Waiting lists and waiting times (P) |
| Understanding Long COVID and its management | Long COVID prevalence (L) |
| | Long COVID unknowns (L; P) |
| | Reluctance to diagnose (P) |
| | Tests/investigations (P) |
| | Being believed (P) |
| | Evolving evidence base (L; S) |
| | Education of PwLC and community rehabilitation staff (S) |
| | Information sharing (S) |
| | Learning from other conditions (S) |
| | Confidence of community rehabilitation staff (S) |
| Long COVID services | Current services strengths and limitations (L; P; S) |
| | Resourcing services (L; P; S) |
| | Need for Long COVID services (L; P; S) |
| | Navigating politics (L) |
| L, Long Covid leads; P, PwLC, People with experience of Long COVID rehabilitation; S, allied health professionals (AHPs). | |

*'[We] need to get past the public's perception that GPs are shut so they return to their GP and get a referral to their service if required.'* [L3102]

All participant groups were aware of the dedicated Long COVID service in HB4. However, due to the publicity the service had received and the proactive development of a clear GP referral pathway, demand for this service quickly exceeded capacity. Staff (AHPs and Long COVID leads) expressed concern that promoting a Long COVID service in their own board may result in the same situation. Their concerns were compounded by the widely acknowledged pressure in the healthcare system due to a range of factors including staff redeployment, absence and reopening of non-essential rehabilitation services that had been closed during previous phases of the pandemic.

*'They [HB4 Long COVID service] were basically inundated. So, I had to wait, about four months or so, something like that for a slot. But it was quicker than I thought, but they were just getting overwhelmed with the workload.'* [P4101]

*'Yeah. So, it's [integrated service] not a widely publicised thing, because I don't know if we could cope as a service with the numbers.'* [S1102]

Patients and staff reported both frustration and understanding around long waiting times for Long COVID rehabilitation:

*'The referral side of it they probably done all the right things, but the timescales were just probably horrific but understandable. I know that in hindsight now so, but it didn't take away the frustrations of it.'* [P3208]

*'Difficult due to capacity. Everywhere so busy and up against it. The pressures on us are really extreme.'* [L3101]

The dedicated service in HB4 was accessed by self-referral and delivered by AHP staff. Self-referral was viewed positively by patients; however, staff reported that the lack of information on self-referrals made appropriate triage of patients challenging. Health boards without self-referral and/or a dedicated Long COVID service also expressed concern that people with Long COVID would not be prioritised for rehabilitation within usual community rehabilitation services.

*'I don't have self-referral at the moment. And the referrals coming through the GP or through something, some of them are coming through. But I don't know where all the others are, so, I'm not blocking access, but they are getting stuck in the system, in my system, in that if I don't have enough information on the referral, they're just going on to the normal routine waiting list.'* [S2207]

HB4 reported the lack of medical staff as a limitation to their dedicated service and, in keeping with other health boards, reported that the lack of a clear Long COVID pathway was a limitation, particularly as people often presented with complex issues that were unsuitable for immediate rehabilitation.

*'But it's those that have the more complex needs that I wouldn't know where to send them to.'* [S2103]

## Understanding Long COVID and its management

Patients commented on the burdensome nature of managing Long COVID and there being a reluctance among GPs to provide a diagnosis. They noted concerns about documenting Long COVID in healthcare records and a need to rule out other underlying conditions.

*'It's quite difficult and just feel like, you know, you have a list as long as your arm when you actually speak to the GP because some things have changed and you know they're very sympathetic, but very clear and honest that well, we don't really know, you know? We just don't know. So, we'll give you this pill to try and treat this symptom at the moment and, that's quite hard.'* [P4101]

The lack of diagnosis was also observed by AHPs and Long COVID leads, who shared a discomfort associated with labelling patients.

*'What are we shaming people for having a condition? What's that all about? It's very bizarre. You get them help. I don't understand. It's like invisible illnesses.'* [L1202]

*'It was really difficult actually to know whether to adopt the term Long COVID 'cause it's not a medical term, but we've decided in the end it's probably got wider recognition, now, that's what the patients are using. But having to be very careful, actually even when you're writing letters to GPs, you know if they didn't have confirmed COVID in the beginning, it's a presumed COVID illness, you know symptoms consistent with Long COVID, you know. It's a lot of kind of working around the houses I feel.'* [S4101]

Most patients expressed a need to be heard and believed by healthcare professionals. They also felt that there was powerful value in hearing that other people with Long COVID experienced similar symptoms, which provided a helpful process in validating patients' unmet needs.

*'It felt to me like it was a bunch of very random symptoms. It didn't seem connected, and I think probably one of the best things in that first conversation was just hearing that the symptoms I have were very common and they are all related.'* [P4102]

*'I get the impression that a lot of my symptoms are replicated a lot across, across other people so and I'm getting a lot of assurance on what people's progress like recovery-wise again its always pinned by keep your expectations to a minimum, like it's no [not] going to be the miracle cure.'* [P3208]

Both AHPs and Long COVID leads spoke of the challenges of understanding Long COVID and its constantly evolving evidence base, as well as the lack of data on prevalence. AHPs also reported a lack of evidence to support best practice for managing Long COVID, which could result in anxiety around patient management.

*'So, what I was concerned about was what if I give the wrong advice or the wrong sort of exercise prescription, and that I*

*actually cause harm to him by something that I've done. And I watched some of the podcasts to sort of understand what we should and shouldn't be doing, and that is when I started to realize that there is a really strong link with Long COVID and the chronic fatigue syndrome and ME population who have been saying for years and years that, actually, sometimes, graded exercise therapy is really harmful, and you absolutely should not prescribe it.'* [S2104]

To increase understanding of Long COVID, AHPs sought out peer support, online resources and knowledge from other conditions to determine how to support people with Long COVID.

*'There's certainly discussions around what is the best way to approach management of Long COVID and there's been a few things we can do, though, because nationally there's a huge focus on physiotherapy management of Long COVID you know, the CSP [Chartered Society of Physiotherapists], I've got lots of stuff out there, but we've also got the clinicians and the COVID rehab team there that although they're not seeing patients and more than happy to signpost people to resources and help with a bit of professional advice. I would say from a Community hub, clinician point of view, confidence hasn't been the main issue. They do seek professional advice on how to manage them.'* [S3203]

### Strengths and limitations of existing Long COVID services

A characteristic of all participating health boards during the study period was the dynamically evolving nature of the services provided. This was partly due to external factors relating to COVID prevalence, the restrictions this placed on service delivery and financial planning decisions. Two health board areas changed their organisational structures affecting services. One health board (HB4) commenced a limited dedicated Long COVID rehabilitation service, delivered by an occupational therapist and a physiotherapist. The service developed clear pathways for referral that gained considerable publicity, as a result of which, the service was unable to meet demand. This was compounded by staff leaving their post. A considerable waiting list developed. The service ran for 18 months after which there was no further funding to continue its delivery, the service ceased and individuals on the waiting list were distributed to local community rehabilitation teams.

### Long Covid service strengths

By round 2 of data collection, AHPs in all health boards were able to see PwLC face-to-face, conduct home visits, have greater integration into the community and begin working towards return-to-work packages with appropriate Long COVID patients. All these service features were perceived by clinicians to increase the flexibility of service delivery. People with Long COVID had experienced a range of modes of service delivery, largely a blended model of face-to-face individual, group or online interventions. However, irrespective of the delivery model, most PwLC had a positive experience of Long COVID rehabilitation, especially the self-management information they received. People with Long COVID reported that validation of their experience was particularly significant.

*'I'm certainly more educated, already daily life has improved I'm not making myself ill doing things I didn't know were doing that.'* [P3203]

*'…the relief to at that point to speak to a professional who was actually acknowledging that this was happening and you know, it wasn't all in my head. I don't know there, for months and months and months I think it was common there was just no-one apart from my GP, there was just no of recognition or acknowledgement of what people were going through or anything like that. So, it was just such a relief to speak to somebody who was just genuinely interested and very supportive.'* [P4107]

In the two health boards where dedicated Long COVID rehabilitation teams existed, AHPs reported that the MDT approach worked well. It was perceived to increase communication, enabling staff to develop both informal learning about Long COVID through peer discussions and also to develop their knowledge and skills in working with the emerging Long COVID clinical population through in-service learning sessions. The Long COVID leads in these areas perceived that the services were valued by people receiving their care and support, which was corroborated by the patient data.

*'It was my GP that referred me to the psychologist then obviously the two of them were interacting so that the [COVID team] were speaking to the psychologist as well and just basically building a care package for me, so it was tailored to what I was needing physically and mentally, so they were quite good actually talking to each other as well, which was I suppose positive.'* [P4105]

AHPs and Long COVID leads from all health boards reported high levels of staff engagement in learning about Long COVID. In the absence of specific evidence-based guidance on Long COVID rehabilitation, AHPs reported drawing on information from a wide variety of sources. Sources included a sparse literature that had been published on rehabilitation, guidance from professional bodies such as the Chartered Society of Physiotherapy and Royal College of Occupational Therapists, and learning from established approaches for conditions with similar presenting features, such as functional neurological disorder, chronic fatigue syndrome and fibromyalgia.

AHPs and Long COVID leads reported that several staff developed an individual professional interest in the condition, worked to increase their knowledge, and sought help to learn from peers locally, across health boards in Scotland, and more widely across the UK.

*'England has been incredibly helpful in giving and sharing lots and…you know… even within the LOCO-RISE study*

*that's been a really helpful kind of network and relationship to support.'* [L3203]

Long COVID leads in health boards where Long COVID rehabilitation services existed reported that these services enabled increased access to care with appropriate referral routes and adaptable and flexible services. One lead described the merit of a blended model of delivery and believed that telephone triage was a useful method of gaining information about patients' needs.

*'Do you know, I think the use of the increased use of digital services has been fantastic. It's meant that we've been able to reach people when we wouldn't have normally been able to reach them. It has to be a blended model. It has to be that those people that need face-to-face intervention get it at a time when it's right. But digital advances have really made a big difference for us.'* [L1101]

### Long COVID service limitations

All participating health boards experienced considerable challenges in delivering Long COVID rehabilitation services. A key challenge was funding, as dedicated funding for Long COVID services had not been allocated by the Scottish Government Long COVID fund at the time these data were collected. The two health boards that had developed dedicated services used existing funding allocations. The short-term nature of this funding in one health board meant that their limited service could not be sustained, and that service ceased during round 2 of our data collection. This limited investment in developing the structure for Long COVID rehabilitation was highlighted by participants as a considerable limiting factor in service delivery, regardless of whether the health board had developed a dedicated service or not, and the *'historical underinvestment in rehab services'* [L2101] was also highlighted by Long COVID leads as a significant limiting factor. Where Long COVID dedicated services had not been developed, services reported that they lacked the staffing capacity to meet the perceived additional service demand.

*I think we're struggling even staffing wise with. We can't staff wards properly. We can't. You know all these things it's. How will we staff a Long COVID service? How will we? How? How will we best provide that service? The NHS way has always been to kind of just absorb it into existing services.'* [S1202]

However, where individuals with Long COVID were seen by existing rehabilitation services these services (described henceforth as 'integrated services') were observed to receive considerably fewer referrals for Long COVID rehabilitation. This appeared to be due to the reluctance in these health board areas to promote services (as discussed above). AHPs in integrated services also reported that their referral criteria were often not appropriate for people with Long COVID, resulting in referrals being rejected as inappropriate as PwLC often

did not meet pre-existing service criteria for community rehabilitation. Fewer Long COVID referrals entering the services led to reduced confidence among AHPs in their ability to appropriately manage PwLC symptoms, with some admitting they lacked confidence due to Long COVID being a new and largely unknown condition for which they had no training.

*'I suppose the other challenge from my point of view is actually just setting up a new service from scratch, so you know that's a big job in itself and setting up without the evidence base without kind of any benchmark you know to work with. And so, trying to do that alongside learning about a new disease you know with information coming out constantly.'* [S4101]

AHPs in integrated services also reported a lack of knowledge and confidence in working with people presenting with Long COVID due to the broad range of symptomatology.

*'Hardest thing—so many different symptoms that you can have with Long COVID.'* [S2204]

Finally, less experienced AHPs in integrated services were reported to lack confidence in dealing with Long COVID cases as they saw fewer PwLC. This meant that in some areas, increased pressure was placed on the limited capacity of senior staff. A limitation within both integrated and dedicated rehabilitation services was the lack of medical input to both integrated and dedicated services and the challenges of onward referral to specialist secondary care.

*'Having the right staff, having medical support in the team, that's been a big gap I feel, and that's still a gap and I think that's really important given the complexity of patients coming through. We're doing a lot of '"safety netting' as AHP's at the moment, but I don't think, you know, that that's our job, necessarily. We've had to take on some of that.'* [S4101]

## DISCUSSION

We explored the perceptions and experiences of community rehabilitation for Long COVID from the perspectives of patients, AHPs delivering community rehabilitation services and NHS staff leading on the Long COVID response in four Scottish health boards. We were particularly interested in exploring the barriers and facilitators to Long COVID rehabilitation experienced by these three groups across the four different health boards. We found several barriers to accessing community rehabilitation for Long COVID across both dedicated and integrated services, although for different reasons, and not only during periods when non-essential services were restricted. We found that the uncertainties of Long COVID, including perceived reluctance of GPs to diagnose, and the constantly evolving evidence base were barriers to accessing rehabilitation services, and to AHPs confidently providing Long COVID rehabilitation. We

also found that lack of or short-term funding, limited pre-COVID-19 funding of community rehabilitation services, and reluctance to promote existing services to the Long COVID population were key operational barriers to Long COVID rehabilitation service provision. Facilitators included flexible services with clear referral pathways, MDT working, sourcing professional information, and support from within and outwith health board areas. While PwLC may experience frustration accessing services and the evidence base for Long COVID rehabilitation is not firmly established, support and information received from an empathetic AHP was highly valued.

Barriers to accessing healthcare services for Long COVID have been reported in previous research focusing on GPs[11 26] and the views of people with Long COVID.[27] We have highlighted that specific barriers to accessing community rehabilitation for PwLC are principally related to unclear referral pathways and demand-capacity issues. Barriers to accessing the physical rehabilitation workforce are not a new phenomenon; Long COVID has arguably brought this previously 'relatively neglected' workforce into the spotlight[28] and highlighted the need for adequate human resourcing of rehabilitation services. This concurs with Baz et al's[11] study of accessing healthcare support for Long COVID in England, which identified staff having to use existing, already stretched, services to support people with Long COVID. Brennan et al[26] reported that GPs (the principal gatekeepers to community rehabilitation services) found referral to community services challenging to negotiate. We identified a complex interplay of factors that impacted on referral from GP to community rehabilitation, including in some health boards a reluctance to promote services due to existing or anticipated capacity issues. Indeed, the dedicated Long COVID service in our study was quickly overwhelmed, demonstrating the real need for Long COVID rehabilitation, which the current epidemiology literature supports,[2 29] and highlighting a capacity issue that needs to be urgently addressed.

There is currently no consensus on the optimal model of service delivery for Long COVID, beyond recommendations for it to be multilevel,[14 17] multiprofessional,[14 30–32] person-centred[30] and adequately resourced.[14] Our findings support these recommendations. Services that are flexible to patients' needs and offer a range of delivery modes were valued by patients and staff. The benefit of MDT working was a key theme throughout our findings, from both staff and patients' perspectives. However, the MDTs in our study were composed of various AHPs and in some cases nurses and psychologists; access to medical specialists, as seen in some Long COVID dedicated MDT services or 'one-stop' clinics[17] was notably lacking in the four participating health boards and identified as a limitation by participants.

The uncertainties around Long COVID, its diagnosis (particularly with the withdrawal of widespread testing), and optimal management have all been previously reported,[33] and are inevitable given the newness of Long COVID as a condition and the as yet emerging evidence base. It is encouraging that research on Long COVID rehabilitation is ongoing. It is vital to continue to explore Long COVID rehabilitation service delivery models, as well as the effectiveness of specific rehabilitation interventions, acknowledging that there is unlikely to be one that fits all contexts or geographical locations.[14]

## Strengths and limitations

This study is the first large-scale study to explore community rehabilitation services for Long COVID in Scotland. Scotland has distinct geographical and health policy landscapes that merit study of Long COVID rehabilitation within its specific context. However, this is also a limiting factor as it potentially impacts the transferability of the study's findings elsewhere. However, the environment is only one of the criteria that should be examined when considering issues of transferability; the study's population and intervention characteristics should also be considered.[34] In these two respects, our findings have greater transferability: PwLC present with similar health needs' internationally[2] and require similar cooperation with service providers; and while differently named, similar dedicated and integrated service models of Long COVID rehabilitation described are found elsewhere.[17 35] Ultimately, it is for the reader to consider how transferable the findings of this study are to their geographical, policy and organisational context.[34] We included the perspectives of those receiving and delivering services, as well as those responsible for their oversight. Data collection took place over two rounds, enabling us to capture some of the shifting practices inevitable with a new condition. Several researchers were involved at all stages of data analysis and interpretation, including people with lived experience of Long COVID. Validity of the study findings was, therefore, enhanced by data (two rounds and three participant groups) and investigator triangulation (multiple researchers involved in analytical decisions).[36 37] The study is not without limitations. We selected the four Scottish health boards based on geographical distinction and service delivery variance, as described in our 2019 national survey.[13] However, by the time, we commenced data collection, approaches to Long COVID rehabilitation had already evolved meaning that there were fewer distinct service delivery differences between the four health boards. Nonetheless, our selected health board areas enabled us to investigate differences across the two principal modes of service delivery (dedicated and integrated) in Scotland at the time of writing. While we are confident we interviewed all key Long COVID leads in each participating health board and had high levels of AHP participation, thereby achieving adequate information power[25] in these two groups, PwLC recruitment was limited by low numbers of people accessing Long COVID rehabilitation services. This study investigated the perspectives of three key stakeholder groups: PwLC, AHP staff delivering services and Long COVID service leads involved in wider strategic and organisational issues.

Other relevant stakeholder's perspectives, such as GPs/family doctors were not included in this paper but will be reported in forthcoming LOCO-RISE publications.

## Implications for practice and research

It is likely that there will be an ongoing need for community rehabilitation to support PwLC. It is clear that a 'one-size-fits-all' model of service delivery is unlikely to be desirable in geographically diverse healthcare environments, such as those found in Scotland. The impact of geographical setting was an observably limiting factor on service delivery, as more remote communities had fewer available rehabilitation staff and those that worked in remote rural communities often worked in isolation. However, the issue of rurality was confounded by included health boards with large remote rural populations also providing an integrated service delivery model of Long COVID rehabilitation. Despite this, our analysis of the data presented in this paper suggests the issue of rurality may be less pivotal than whether health boards decide to integrate Long COVID rehabilitation into existing services or develop new dedicated Long COVID rehabilitation services. Regardless of which approach is taken, this study has found several factors that require to be addressed to enable optimal Long COVID rehabilitation service delivery: proactive provision and promotion of clear referral pathways so both PwLC and primary care clinicians know of their existence and how to refer to them; sufficient staffing and range of multidisciplinary expertise to meet the broad range of rehabilitation needs that PwLC require; where health boards provide services in remote and rural areas, AHP staff should be enabled to access professional information and specific peer support from clinicians within and outwith their own Health Board area who have greater experience of and confidence in providing rehabilitation to PwLC. This study represents the first stage in exploring community rehabilitation for Long COVID. We have also explored the perceptions of GPs and adults with Long COVID who have tried to access rehabilitation services (to be reported elsewhere). This paper is part of LOCO-RISE, a wider realist evaluation of Long COVID rehabilitation services in the four health boards in Scotland. Future papers will present further data from this evaluation. Collectively these papers will inform contextualised, evidence-based recommendations for future Long COVID rehabilitation services in Scotland. The transferability of key findings from this Scottish-based study should be considered when considering their implications for different clinical and policy contexts.

## CONCLUSION

The organisational delivery of Long COVID community rehabilitation is complex and faces several challenges. In this first of a series of outputs from our LOCO-RISE study, we have provided a detailed understanding of the barriers and facilitators to Long COVID community rehabilitation from the perspectives of patients, AHPs and Long COVID leads. The knowledge presented here can be used by those developing and delivering services for people with Long COVID. Community rehabilitation services have much to offer people with Long COVID, but their optimal delivery requires adequate planning, publicity and resource.

**Author affiliations**
[1]NMAHP-RU, University of Stirling, Stirling, UK
[2]Scottish Centre for Evidence-Based Multi-Professional Practice, Aberdeen, UK
[3]School of Health Sciences, Robert Gordon University, Aberdeen, UK
[4]NMAHP Research Unit, Glasgow Caledonian University, Glasgow, UK
[5]Robert Gordon University, Aberdeen, UK
[6]SNHS, University of Dundee, Dundee, UK
[7]Healthcare Improvement Scotland, Edinburgh, UK
[8]Douglas Grant Rehabilitation Centre, Ayrshire Central Hospital, Irvine, UK

**Acknowledgements** We are grateful to all our participants who gave their time to take part in interviews and to the local primary investigators from the health boards; Lynne Frew, Lynn Morrison and Gail Thomson-Patel, for their contributions to the LOCO-RISE study. We extend our sincere gratitude to the rehabilitation and administrative staff for supporting recruitment and data collection, and to Valery Burnett (VB) for her assistance with data collection.

**Contributors** KC, ED, JHM, LA, JP, JO and AL contributed to the study's conception and design. TT, JC, RM and JS undertook data collection. ED, KC, TT, JC, JS, ES and RM undertook analyses and drafted the first version of the manuscript. ED led future iterations of the manuscript. All authors read and commented on the manuscript and approved the final version of it. The corresponding author attests that all authors meet authorship criteria and that nobody meeting the criteria have been omitted. ED and KC are joint lead authors and guarantors. They had joint responsibility for the research conduct of the study, had access to the data and controlled the decision to publish.

**Funding** This work was supported by the Chief Scientist Office Scotland, grant number COV/LTE/20/29.

**Competing interests** All authors have completed the ICMJE uniform disclosure form at www.icmje.org/coi_disclosure.pdf and declare: all authors had financial support from the Chief Scientist Office Scotland (grant number COV/LTE/20/29) for the submitted work; ED has received research grants from NIHR and Scottish Government for the following studies: Caring for Long COVID in primary care and DBI COVID study, respectively; no other relationships or activities that could appear to have influenced the submitted work.

**Patient and public involvement** Patients and/or the public were involved in the design, or conduct, or reporting, or dissemination plans of this research. Refer to the Methods section for further details.

**Patient consent for publication** Consent obtained directly from patient(s).

**Ethics approval** This study involves human participants and ethical approval was granted from the Wales Research Ethics Committee 6 (21/WA/0118), and each of the four health boards granted R&D management approval. Participants gave informed consent to participate in the study before taking part.

**Provenance and peer review** Not commissioned; externally peer reviewed.

**Data availability statement** Data are available on reasonable request. All data produced in the present study are available on reasonable request. Robert Gordon University holds the copyright for the full interview transcripts and may grant data sharing permission on request.

**ORCID iDs**
Edward Duncan http://orcid.org/0000-0002-3400-905X
Lyndsay Alexander http://orcid.org/0000-0003-2437-9150
Julie Cowie http://orcid.org/0000-0002-4653-1283
Jacqui H Morris http://orcid.org/0000-0002-9130-686X
Joanna Shim http://orcid.org/0000-0001-9438-9640
Emma Stage http://orcid.org/0009-0008-5309-8116
Kay Cooper http://orcid.org/0000-0001-9958-2511

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
