## [Reviewer comments · BMJ Open]

ARTICLE DETAILS

TITLE (PROVISIONAL)	Investigating Scottish Long COVID community rehabilitation service models from the perspectives of people living with Long COVID and healthcare professionals: a qualitative descriptive study
AUTHORS	Duncan, Edward; Alexander, L; Cowie, Julie; Love, Alison; Morris, Jacqui H.; Moss, Rachel; Ormerod, Jane; Preston, Jenny; Shim, Joanna; Stage, Emma; Tooman, Tricia; Cooper, K

VERSION 1 – REVIEW

REVIEWER	Hatcher, Simon Ottawa Hospital Research Institute
REVIEW RETURNED	01-Sep-2023

GENERAL COMMENTS	Thank you for asking me to review this paper on the perceptions and experiences of different Long Covid rehabilitation service models. This is a qualitative study of using 51 interviews of 8 Long Covid leads, 15 allied health professionals and 15 people with experience of rehabilitation for Long Covid. The study concludes “organisational delivery of Long Covid rehabilitation is complex and presents multiple challenges”. The paper is generally well written. The knowledge gap is unclear. As the authors state there are already “well documented barriers accessing healthcare services” for people with Long Covid. The stated aims were to explore the perceptions and experiences of i) the barriers and ii) the facilitators to Long Covid rehabilitation services. This question was unclear to me – does it mean barriers and facilitators to referral? Or staying in treatment? Or something else? I was unclear why another study was needed given the existing literature. Methodologically I was unsure why the authors chose to use “interview topic guides” to inform their interviews. (These were not included in the Supplemental files of the submission – Supplemental file 1 is the COREQ checklist). The Consolidated Framework for Implementation Research guide was specifically designed to answer the sort of questions that the authors asked. The authors should justify why they didn’t use this approach. I was also unclear why they used a convenience sample of people with Long Covid. The authors didn’t seek out people who may be underrepresented. I understand all this group had been successfully referred to Long Covid rehabilitation. It might have been more informative to seek out people with Long Covid who
--

	hadn't been able to get seen or who were on waiting lists to be seen. I was also surprised to see that family doctors were not interviewed given that the point of the study was looking at barriers and facilitators to accessing Long Covid clinics (or at least I think access was one of the questions the authors were interested in). There is also the issue of generalisability. As the authors state the provision of services for Long Covid is evolving. Drawing conclusions from a study done over a year ago in one particular location limits the generalisability of this study to potential readers. I found its conclusions underwhelming and I'm not sure how they would help anyone design or implement Long Covid rehabilitation services.
--	--

REVIEWER	Paranjape, Swati King Edward Memorial Hospital, Department of Physiotherapy
REVIEW RETURNED	12-Sep-2023

GENERAL COMMENTS	This is a very interesting and elaborate study using qualitative inquiry. There are several strengths of this study e.g. inclusion of country based health boards with geographical and demographic spread, which gave more in-depth description from all the stakeholders. However certain areas as below if elaborated can give more clarity to readers.  1. Methodological decisions with explicit identification of rationale to use this research design and framework analytical approach over other qualitative approaches 2. Sampling frame / exclusions/ data saturation/ data validation or triangulation strategies 3. Temporal and geographical (rural or urban) effect on quality, and accessibility of services and perceptions of stakeholders; as that was a deliberation in this study 4. Rationale of use of TIDieR intervention description categories (as there is no intervention in this study) 5. Methodological/ analytical limitations of this study Some questions: With respect to analysis, if research team was diverse how were researchers' biases taken care of ? AHPs were also from different professions hence, their perceptions would certainly differ while handling Long COVID. How was that analyzed? With respect to quote on page 16 (line 6 and line 17-19) was inclusion of organizational representation thought of to get more holistic picture? Wishing Bests
---

VERSION 1 – AUTHOR RESPONSE

Reviewer 1		
1. The knowledge gap is unclear. As the authors state there are already “well	1. We were exploring community rehabilitations services, rather	2

documented barriers accessing healthcare services” for people with Long Covid. The stated aims were to explore the perceptions and experiences of i) the barriers and ii) the facilitators to Long Covid rehabilitation services. This question was unclear to me – does it mean barriers and facilitators to referral? Or staying in treatment? Or something else? I was unclear why another study was needed given the existing literature. 2. Methodologically I was unsure why the authors chose to use “interview topic guides” to inform their interviews. (These were not included in the Supplemental files of the submission – Supplemental file 1 is the COREQ checklist). The Consolidated Framework for Implementation Research guide was specifically designed to answer the sort of questions that the authors asked. The authors should justify why they didn’t use this approach. 3. I was also unclear why they used a convenience sample of people with Long Covid. The authors didn’t seek out people who may be underrepresented. I understand all this group had been successfully referred to Long Covid rehabilitation. It might have been more informative to seek out people with Long Covid	than healthcare services in general. Rehabilitation has not been the focus of previous research – we have clarified this in the background. Regarding the question, we have provided more detail at the end of the background – we were exploring barriers & facilitators to implementation of Long COVID rehabilitation, therefore access, delivery & receipt of Long COVID rehabilitation were all being evaluated. 2. The following text has been added to page 6: The topic guides combined a theory-based approach using the TIDieR intervention description categories [21] to enable a comprehensive description of each rehabilitation service, questions to long COVID leads based on constructs from the Consolidated Framework for Implementation Research (Ref) to ascertain the factors that were associated with the intervention’s implementation and a series of questions exploring facilitators and barriers to delivering and receiving community Long COVID rehabilitation. The interview topic guides were pilot tested with people who had lived experienced of Long Covid and with two AHPs who were not involved in the study prior to implementation. 3. The aim was to explore barriers & facilitators to implementing Long COVID rehabilitation (see response to point 1 above). We therefore recruited from Long COVID rehabilitation services and explored PwLC experiences of accessing and receiving rehabilitation. A key finding was the apparent barriers to PwLC being referred for rehabilitation.	3 6 8 8
--	---	-------------------------------------

who hadn't been able to get seen or who were on waiting lists to be seen. 4. I was also surprised to see that family doctors were not interviewed given that the point of the study was looking at barriers and facilitators to accessing Long Covid clinics (or at least I think access was one of the questions the authors were interested in). 5. There is also the issue of generalisability. As the authors state the provision of services for Long Covid is evolving. Drawing conclusions from a study done over a year ago in one particular location limits the generalisability of this study to potential readers. 6. I found its conclusions underwhelming and I'm not sure how they would help anyone design or implement Long Covid rehabilitation services.	Although not reported in this study, we subsequently conducted a related study to explore this phenomenon in more detail – the findings of which will be reported separately. See comment page 8. 4. As explained above (point 3) we were exploring the implementation of Long COVID rehabilitation. One of the key barriers to emerge was that of access, despite our prior survey informing us that pathways existed in all 4 Health Boards taking part in the study. This has led to a sperate piece of work exploring GPs and community-dwelling PwLC experiences of accessing/trying to access community rehabilitation for Long COVID (manuscript in preparation). 5. The study is limited in terms of generalisability, as discussed in 'implications for practice': "The transferability of key findings from this Scottish -based study should be considered when considering their implications for different clinical and policy contexts." However, we assert that although the study was conducted over a year ago, the findings are relevant as Long COVID services are still developing in Scotland, and beyond, and will perhaps become even more relevant as services need to consider how to respond to short-term Government funding coming to an end. We have expanded the section on research and practice implications to provide further clarity of what this paper contributes and where it sits as the first of a series of publications from our larger study. The key findings regarding improved clarity of referral pathways and	
---	--	--

	increased support for services working in remote and rural communities have already been taken on board by policy makers in the Scottish Government.	
Reviewer 2		
1. Methodological decisions with explicit identification of rationale to use this research design and framework analytical approach over other qualitative approaches	1. We have added a sentence to study design justifying the use of QD and to data analysis explaining that Framework is congruent with the QD design.	3, 7
2. Sampling frame / exclusions/ data saturation/ data validation or triangulation strategies	2. Please see response to Editor (point 3) regarding sampling frame & saturation (information power). We have added information on exclusion criteria for each participant group. We have added a comment about triangulation/validity of findings to the 'strengths & limitations' section.	6,7,20 6 20
3. Temporal and geographical (rural or urban) effect on quality, and accessibility of services and perceptions of stakeholders; as that was a deliberation in this study	3. We have added further information on the geographical effect of health boards on service delivery in the implications setting of the revised paper.	20-21 6
4. Rationale of use of TIDieR intervention description categories (as there is no intervention in this study)	4. The TIDieR framework was used to enable a comprehensive description of each rehabilitation service. In this context the rehabilitation service model can be conceived of as a service level intervention.	19-20
5. Methodological/ analytical limitations of this study	5. We have expanded the limitations section to include the lack of General Practitioners/family doctors and described how we sought "information power" rather than data saturation.	
6. With respect to analysis, if research team was diverse how were researchers' biases taken care of ?	6. Please see page 7, where we have added text to justify the use of multiple researchers from different disciplinary backgrounds and the mitigation of potential bias.	
7. AHPs were also from different professions hence, their perceptions would certainly differ while handling		

Long COVID. How was that analyzed? 8. With respect to quote on page 16 (line 6 and line 17-19) was inclusion of organizational representation thought of to get more holistic picture?	7. Our prior knowledge of the included services informed us that they were multidisciplinary in nature. Long COVID rehabilitation (as our findings demonstrate) can be delivered by a range of AHPs, but predominantly OT and physio, often working together. We did not find anything in the data that suggested different perceptions by profession, therefore we chose to analyse by the umbrella grouping of “AHP”. 8. The inclusion of Long Covid Leads was to gain an understanding of the wider strategic challenges of delivering a long covid rehabilitation service. The distinct contributions of each participant group have been more fully explained in the study design section of the revised submission.	
--	---	--

Please read and respond to all of the peer review comments. You should provide a point-by-point response to explain any changes you have (or have not) made to the original article and be as specific as possible in your responses.

The original files will be available to you when you start your revision. Please delete any files that you intend to replace with updated versions and upload the following using the appropriate file designation:

- “Main Document” - This is a clean copy (without tracked or highlighted changes) of your revised article. Please delete your original submission file.
- “Main Document - marked copy” - This is the edited version of your original article, including edits to address the peer review comments. Any changes have been highlighted using a track change function or bold or coloured text.

Please replace any other files that have been updated e.g. Images, forms.

The reviewers' comments, your response, and the previous versions of your article will be published as supplementary information alongside the final version of your article.

Information relating to your article, including author names and affiliations, title, abstract and required statements (e.g. competing interests, contributorship, funding) will be taken directly from the information held in ScholarOne, and not from the article file. Please check that this information has been entered correctly and has been updated as appropriate. If your revised article is accepted, you will only be able to make minor changes (e.g. correction of typesetting errors and proof stage) prior to publication.

Please submit your revised article by 14-Oct-2023. If you have not submitted by this date, or would like an extension, please email our Editorial Office.

VERSION 2 – REVIEW

REVIEWER	Hatcher, Simon Ottawa Hospital Research Institute
REVIEW RETURNED	08-Nov-2023

GENERAL COMMENTS	Thank you for sending me a revised manuscript of this study. The authors have gone some way in addressing the questions raised in my first review. However it still lacks generalisability - even within Scotland as the authors state provision of Long Covid rehabilitation had already evolved by the time they started data collection. Second the results are still generic and could describe difficulties in setting up any rehabilitation service (or indeed any clinical service). I am still not sure how the results could help with planning Long Covid rehabilitation services.
--

REVIEWER	Paranjape, Swati King Edward Memorial Hospital, Department of Physiotherapy
REVIEW RETURNED	03-Nov-2023

GENERAL COMMENTS	The revised manuscript addresses most of the questions raised in the first review. However, there are few new revisions needed. Following comments may help revising the manuscript further.  1. Title: Authors have changed the previous title which now mentions the study as 'mixed methods case study design'. However, study employs only qualitative enquiry and not a Mixed Method Design. "Case Study" in qualitative enquiry design essentially refers to the case or cases at hand or to be studied which are novel, different, unusual or complex in context. Authors need to understand the concept of "Mixed Method Research Designs" and 'Qualitative Designs'. This is major flaw of this revision. In abstract authors mention it as 'qualitative descriptive design' and stick to that all throughout the text body. Checklist also refers to qualitative enquiry. Rectifications are needed in the title. Title uses the word 'understanding'. If authors feel it may be modified as 'exploring' 'studying'. 2. Limitations: The present version revises "Limitations" section however, it essentially should refer to limitations in terms of study design, generalizability, applications/implications of study findings, or any other in contextual terms. If that also can be added it would be more balanced report. 3. Introduction: It mentions about a larger project LOCO-RISE. It would be appropriate and would offer clarity to readers if abbreviations are explained or brief about the project is offered (since this paper is a part of that project) 4. Recruitment: This being qualitative study identifying a set criteria of sample (sample size) is essential. What was the criteria of selecting the given number of AHPs, Long COVID Leads and Patients? Elsewhere text mentions that instead of 'saturation' 'information power' was used. That needs to be clearly defined. Rest all sections are appropriately written and discussed. Wishing Bests
---

VERSION 2 – AUTHOR RESPONSE

Reviewer 1		
1. Title: Authors have changed the previous title which now mentions the study as 'mixed methods case study design'. However, study employs only qualitative enquiry and not a Mixed Method Design. "Case Study" in qualitative enquiry design essentially refers to the case or cases at hand or to be studied which are novel, different, unusual or complex in context. Authors need to understand the concept of "Mixed Method Research Designs" and 'Qualitative Designs'. This is major flaw of this revision. In abstract authors mention it as 'qualitative descriptive design' and stick to that all throughout the text body. Checklist also refers to qualitative enquiry. Rectifications are needed in the title. Title uses the word 'understanding'. If authors feel it may be modified as 'exploring' 'studying'. 2. Limitations: The present version revises "Limitations" section however, it essentially should refer to limitations in terms of study design, generalizability, applications/implications of study findings, or any other in contextual terms. If that also can be added it would be more balanced report. 3. Introduction: It mentions about a larger project LOCO-RISE. It would be appropriate and would offer clarity to readers if abbreviations are explained or brief about the project is offered (since this paper is a part of that project). 4. Recruitment: This being qualitative study identifying a set	1. As requested, we have changed the title so the design referred to in the title reflects the qualitative descriptive design we used and refer to throughout the paper. Apologies for that accidental inconsistency. We have also changed the word 'Understanding' in the title to 'Investigating', which we trust is sufficiently in line with the reviewer's wishes. 2. Thank you for this suggestion. We have considerably expanded the Recruitment and Data Collection section and the Strengths and Limitations section to address issues of Generalisability and the study's application/implications (These issues in a qualitative design are more appropriately considered as relating to the construct of Transferability and we have referred to it as such in the paper). 3. We have added further information about LOCO-RISE, as requested. 4. We have added further information about how we decided our sample size and provided further information to clarify the construct of "information power" and how it relates to our sample, as requested.	1 6; 19-20 3 6; 19-20

criteria of sample (sample size) is essential. What was the criteria of selecting the given number of AHPs, Long COVID Leads and Patients? Elsewhere text mentions that instead of 'saturation' 'information power' was used. That needs to be clearly defined.		
Reviewer 2		
1. The authors have gone some way in addressing the questions raised in my first review. However it still lacks generalisability - even within Scotland as the authors state provision of Long Covid rehabilitation had already evolved by the time they started data collection. Second the results are still generic and could describe difficulties in setting up any rehabilitation service (or indeed any clinical service). I am still not sure how the results could help with planning Long Covid	 1. See comment above relating to issues of Generalisability. We have added further information in our paper regarding the strengths and limitations of the transferability of our study findings to other settings. 2. Issues relating to access and delivery of Long COVID rehabilitation services are currently very topical in the UK and beyond as people with Long COVID continue to struggle to access appropriate rehabilitation services and services struggle to be adequately and appropriately developed. The facilitators and barriers to such service development outlined in this study have been welcomed in informal presentations of our key study findings to national health care policy makers and implementers who await formal publication of this paper with interest. Furthermore, this paper is the first in a series of papers that will address this issue and cumulatively support service planning for Long COVID rehabilitation. 	19-20

VERSION 3 – REVIEW

REVIEWER	Paranjape, Swati King Edward Memorial Hospital, Department of Physiotherapy
REVIEW RETURNED	14-Nov-2023

GENERAL COMMENTS	Now the paper is well balanced and appropriate for publication. Wishing Bests
--